# ETAB: A Benchmark Suite for Visual Representation Learning in Echocardiography

**Ahmed M. Alaa**
UC Berkeley and UCSF
Berkeley, CA
amalaa@berkeley.edu

**Anthony Philippakis**
Broad Institute of MIT and Harvard
Cambridge, MA

**David Sontag**
MIT CSAIL and IMES
Cambridge, MA
dsontag@csail.mit.edu

## Abstract

Echocardiography is one of the most commonly used diagnostic imaging modalities in cardiology. Application of deep learning models to echocardiograms can enable automated identification of cardiac structures, estimation of cardiac function, and prediction of clinical outcomes. However, a major hindrance to realizing the full potential of deep learning is the lack of large-scale, fully curated and annotated data sets required for supervised training. High-quality pre-trained representations that can transfer useful visual features of echocardiograms to downstream tasks can help adapt deep learning models to new setups using fewer examples. In this paper, we design a suite of benchmarks that can be used to pre-train and evaluate echocardiographic representations with respect to various clinically-relevant tasks using publicly accessible data sets. In addition, we develop a unified evaluation protocol—which we call the *echocardiographic task adaptation benchmark* (ETAB)—that measures how well a visual representation of echocardiograms generalizes to common downstream tasks of interest. We use our benchmarking framework to evaluate state-of-the-art vision modeling pipelines. We envision that our standardized, publicly accessible benchmarks would encourage future research and expedite progress in applying deep learning to high-impact problems in cardiovascular medicine.

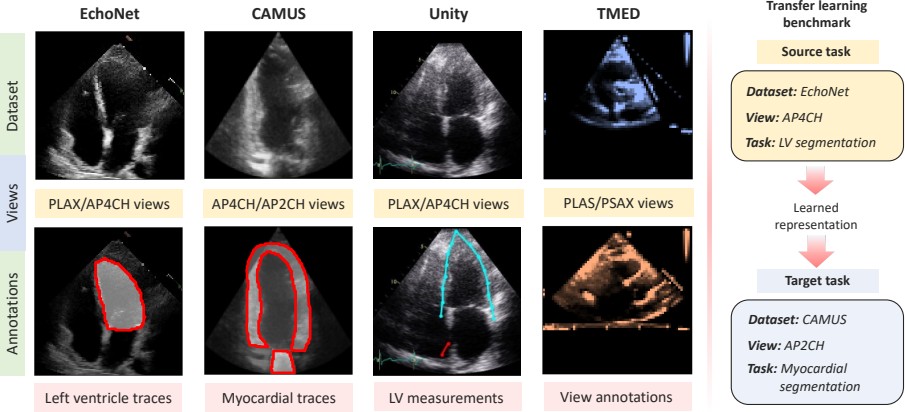

Figure 1: **Visual task adaptation for echocardiography.** Using multiple open- and public-access echocardiogram data sets, we develop a suite of benchmarks for representation and transfer learning in echocardiography.

36th Conference on Neural Information Processing Systems (NeurIPS 2022) Track on Datasets and Benchmarks.

# 1    Introduction and Motivation

Echocardiography is one of the most commonly used non-invasive imaging techniques for assessing cardiac function, examining heart anatomy and diagnosing cardiovascular diseases. An "echo" study is an ultrasound of the heart acquired by a cardiac sonographer through a transducer device—different acquisition angles by which the device is placed relative to the patient's heart provide different "views" of the heart anatomy. Echocardiograms inform cardiologists, surgeons, oncologists and emergency physicians on clinical decisions pertaining to treatment management and surgical planning [1–6]. Because of the central role it plays in cardiovascular medicine, there has been a significant interest in applying deep learning-based computer vision models to echocardiograms, with the ultimate goal of automating cardiac evaluation, reducing variance and improving reproducibility in interpreting echocardiograms, and predicting patient-specific clinical outcomes [7–14].

A key hindrance to wider engagement in research focusing on applying deep learning to echocardiography is the lack of large, standardized, publicly-accessible data sets with the annotations required for all downstream tasks of interest. This is because, in addition to the regulatory hurdles associated with publicly sharing clinical data, building such data sets is already a difficult task in itself—i.e., collecting all labels of interest would require examination of clinical reports, pre-selection of relevant echo views, and retrospective manual annotation by experts. Consequently, existing public data sets for echocardiography (e.g., [15–17]) tend to contain a modest number of samples, with only subsets of the annotations and views required for all possible downstream analyses.

Because many downstream tasks would share the same relevant echocardiographic features, access to high-quality representations of echocardiograms—i.e., representations that retain these features—could enable reusing the same pre-trained representations across many tasks rather than training a new model from scratch for each task [18–20], hence enabling adaptation of pre-trained models in setups where fewer annotated data might be available. We shall call such a procedure *visual task adaptation*—see Figure 1 for a pictorial illustration. Motivated by such thinking, this paper develops a standardized set of benchmark tasks for pre-training and/or evaluating visual models of echocardiography. Our goal is to increase engagement of researchers in a high-impact application domain by providing a systematic domain-specific benchmark through which researchers can focus their effort on tackling challenging problems peculiar to echocardiography, and translate methodological advances in representation learning to practical problems in cardiovascular medicine.

**The contribution of this paper is two-fold.** *First*, we develop a comprehensive suite of benchmark tasks tailored to echocardiography using a meta-dataset of public-access sources of echocardiographic data. These benchmark tasks can be used to test vision modeling pipelines, pre-train visual representations of echoes, and evaluate the transferability of representations across pairs of source-target tasks which might differ in the data source, echo views and annotations. *Second*, using our suite of benchmark tasks we specify a unified evaluation protocol for readily available pre-trained representations—the *echocardiographic task adaptation benchmark* (ETAB)—which is meant to evaluate the extent by which a given representation generalizes to different tasks, views and patient cohorts. The ETAB benchmark enables researchers to share a unified and publicly-accessible evaluation protocol, even when the representations themselves are pre-trained on private hospital data. ETAB is implemented as a full-fledged software package with a user-friendly API for researchers to design and evaluate visual representations for echocardiograms (Code available at: https://github.com/ahmedmalaa/ETAB).

# 2    Echocardiogram Datasets

The ETAB benchmark suite is based on 5 publicly accessible echocardiogram data sets that span different cohorts and involve different echocardiographic views and annotations (Table 1). In what follows, we provide a brief overview of all data sets currently supported in ETAB.

**EchoNet.** This data set has two variants: EchoNet-Dynamic and EchoNet-LVH. In the EchoNet-Dynamic data set, introduced in [15], one apical-4 chamber (AP4CH) 2D gray-scale video is extracted from each echo study. A total of 10,036 videos are collected from 10,036 distinct individuals who underwent echocardiography between 2006 and 2018 as part of routine care at a University Hospital. Individuals in the data set were selected at random from hospital records. Along with each video, cardiac function assessments and calculations obtained by a registered sonographer and verified by a level-3 echocardiographer are provided. The second data set, EchoNet-LVH, was collected for a

| Data set | Sample size | Echo views | Annotations |
|---|---|---|---|
| **EchoNet-Dynamic** | 10,036 | AP4CH | LV traces and LV ejection fraction (EF) |
| **EchoNet-LVH** | 12,000 | PLAX | LV internal dimension, LV posterior wall, Intraventricular septum |
| **Unity** | 1,224 | AP4CH, PLAX | LV measurements and longitudinal strain |
| **CAMUS** | 500 | AP4CH, AP2CH | LV EF, ES/ED frames, LV epicardium, endocardium and left atrium segments |
| **TMED** | 2,341 | PLAX, PSAX, Other | Aortic Stenosis diagnoses, view annotations |

Table 1: **Publicly accessible data sets involved in our ETAB framework.**

similar cohort and comprises detailed measurements of the LV dimensions based on parasternal long axis (PLAX) views. Both data sets were released by Stanford AIMI Center and are accessible via: https://echonet.github.io/echoNet/ and https://echonet.github.io/lvh/.

**Unity Imaging Collaborative.** These echocardiograms were obtained from the British echocardiography laboratories and were retrospectively annotated by echocardiography-certified cardiologists through a UK-wide initiative that involved 17 hospitals [21]. The data set comprises a mixture of apical and parasternal views; the former is accompanied with measures of longitudinal strain, and the latter involves measures of LV dimensions. The data is accessible via: https://data.unityimaging.net/.

**CAMUS.** The Cardiac Acquisitions for Multi-structure Ultrasound Segmentation (CAMUS) is an open-access data set from 500 patients, acquired at the University Hospital of St. Etienne (France) [16]. Compared to EchoNet, this data set has a smaller sample size but is more elaborately annotated, hence it serves as an appropriate target data set for a variety of downstream tasks. The data set contains apical two- and four-chamber acquisitions (AP2CH and AP4CH). Half of the population has a left ventricle (LV) ejection fraction lower than 45%, thus being considered at pathological risk. The data set contains full annotation of the left atrium, the endocardium and epicardium borders of the LV, performed by a cardiologist. To identify the beginning and end of each cardiac cycle, ED and ES frames within each echo clip are labeled. Data is accessible via: https://www.creatis.insa-lyon.fr/Challenge/camus/.

**TMED.** The Tufts Medical Echocardiogram Dataset (TMED) contains still echocardiogram imagery acquired in the course of routine care from 2015 to 2020 [17]. This data set contains both labeled and unlabeled images—labels include a categorization of views into parasternal long and short axis views (PLAX and PSAX), and diagnostic severity ratings for aortic stenosis (AS). Unlike EchoNet, Unity and CAMUS, this data set contains only still images rather than sequences of frames. The data set is publicly available and can be accessed through: https://tmed.cs.tufts.edu/.

## 3 A Benchmark Suite for Echocardiographic Representation Learning

Our benchmark suite encapsulates a set of standardized tasks that cover a wide variety of echocardiographic modeling problems of interest. Each task comprises a selection of a data set, a view and an annotation from the options listed in Table 1. These standardized tasks can be used to benchmark the performance of vision models, pre-train visual representations, or evaluate the quality of externally pre-trained representations of echocardiograms. In Section 3.1, we present the ETAB benchmark tasks, and in Section 3.2 we describe our proposed ETAB evaluation protocol.

### 3.1 The ETAB benchmark tasks

Based on the data sets listed in Section 2, we design **31 benchmark tasks** that cover various echocardiographic tasks of interest. Each benchmark task comprises a specification of the data set, echo view and annotations. The benchmarks are divided into four categories: (●) cardiac structure identification (e.g., detecting the borders of the LV), (●) cardiac function estimation (e.g., estimating the LV ejection fraction), (●) view classification, and (●) clinical prediction. Benchmarks within the same category share a color code that will be later used in visualizations of experimental results. These benchmark tasks constitute the core of the ETAB evaluation protocol presented later in Section 3.2.

|  | Benchmark description | | | |
|---|---|---|---|---|
|  | **Benchmark code** | Data set | View | Task |
| **Cardiac structure identification** | ● `a0-A4-E` | EchoNet-Dynamic | AP4CH | Left ventricle segmentation |
|  | ● `a0-A4-C` | CAMUS | AP4CH | Left ventricle segmentation |
|  | ● `a1-A4-C` | CAMUS | AP4CH | Left atrium segmentation |
|  | ● `a2-A4-C` | CAMUS | AP4CH | Myocardial wall segmentation |
|  | ● `a0-A2-C` | CAMUS | AP2CH | Left ventricle segmentation |
|  | ● `a1-A2-C` | CAMUS | AP2CH | Left atrium segmentation |
|  | ● `a2-A2-C` | CAMUS | AP2CH | Myocardial wall segmentation |
| **Cardiac function estimation** | ● `b0-A4-E` | EchoNet-Dynamic | AP4CH | LV ejection fraction estimation |
|  | ● `b1-A4-E` | EchoNet-Dynamic | AP4CH | Cardiac cycle phase estimation |
|  | ● `b2-PL-E` | EchoNet-LVH | PLAX | Intraventricular septum estimation |
|  | ● `b3-PL-E` | EchoNet-LVH | PLAX | LV internal dimension estimation |
|  | ● `b4-PL-E` | EchoNet-LVH | PLAX | LV posterior wall identification |
|  | ● `b0-A4-C` | CAMUS | AP4CH | LV ejection fraction estimation |
|  | ● `b0-A2-C` | CAMUS | AP2CH | LV ejection fraction estimation |
|  | ● `b1-A4-C` | CAMUS | AP4CH | Cardiac cycle phase estimation |
|  | ● `b1-A2-C` | CAMUS | AP2CH | Cardiac cycle phase estimation |
|  | ● `b5-A4-U` | Unity | AP4CH | Identifying mitral hinge point on septum |
|  | ● `b6-A4-U` | Unity | AP4CH | Identifying mitral hinge point on lateral wall |
|  | ● `b7-A4-U` | Unity | AP4CH | Identifying LV endocardial apex |
|  | ● `b8-A4-U` | Unity | AP4CH | Identifying LV endocardial contour |
|  | ● `b2-PL-U` | Unity | PLAX | Intraventricular septum estimation |
|  | ● `b3-PL-U` | Unity | PLAX | LV internal dimension estimation |
|  | ● `b4-PL-U` | Unity | PLAX | LV posterior wall identification |
| **View recognition** | ● `c1-XX-E` | EchoNet | — | Classify AP4CH and PLAX views |
|  | ● `c0-XX-C` | CAMUS | — | Classify AP4CH and AP2CH views |
|  | ● `c1-XX-U` | Unity | — | Classify AP4CH and PLAX views |
|  | ● `c2-XX-T` | TMED | — | Classify PLAX and PSAX views |
| **Clinical prediction** | ● `d0-A4-E` | EchoNet-Dynamic | AP4CH | Diagnose cardiomyopathy |
|  | ● `d1-PL-T` | TMED | PLAX | Diagnose Aortic Stenosis |
|  | ● `d0-A4-C` | CAMUS | AP4CH | Diagnose cardiomyopathy |
|  | ● `d0-A2-C` | CAMUS | AP2CH | Diagnose cardiomyopathy |

Table 2: **List of all ETAB benchmark tasks.** The tasks cover four categories: (●) cardiac structure identification, (●) cardiac function estimation, (●) view classification, and (●) clinical prediction. Each benchmark task is assigned a code that encodes a specification of the source data set, echocardiographic view and annotations.

Each benchmark is encoded through an 5-character code of the form XX-XX-X. The encoding scheme describes each benchmark in terms of the data source, echo view and modeling task follows:

- **Characters 0-1:** Task identifier
- **Characters 2-3:** View identifier
- **Characters 4-5:** Dataset identifier

The data sets are encoded as follows **E**: EchoNet, **C**: CAMUS, **U**: Unity, and **T**: TMED. The views are encoded as follows: **A2**: Apical 2-chamber, **A4**: Apical 4-chamber, **PL**: parasternal long axis, and **PS**: parasternal short axis. The 4 task categories are encoded as **a**, **b**, **c** and **d**, and the detailed list of codes for all tasks is provided in the Appendix. The task encoding provides a systematic way for running benchmark experiments and reporting results through the ETAB software library.

**Transfer learning benchmarks.** In addition to the core benchmark tasks listed in Table 2, benchmark setups for transfer learning can be constructed by picking pairs of source and target tasks from the ETAB task suite. The pairs of benchmark tasks would be typically selected so that at least one of the three elements of a task specification (i.e., data set, view or annotation) differs between the source and the target. These "adaptation" setups can be used to test the ability of a transfer learning algorithm

to repurpose pre-trained representations for new downstream tasks, and provide insights into which tasks and model architectures lead to the most generalizable representations of echocardiograms. A transfer learning benchmark is encoded as: `Source-task-code/Target-task-code`. For example, benchmark `a0-A4-E/b3-PL-U` involves pre-training a representation to segment the left ventricle using apical 4-chamber studies from EchoNet-Dynamic, and then using the resulting representation to estimate the LV internal dimension using PLAX studies from the unity data set.

**Vision modeling pipelines.** Each of the core benchmark tasks in Table 2 can be conducted using a *vision modeling pipeline* $\mathcal{M}$ that comprises **(1)** a choice of a backbone architecture for the visual representation and a **(2)** task-specific model head as depicted in Figure 2. The backbone representation can either be pretrained using external data sources or trained from scratch on the benchmark task.

When applied to a transfer learning setup (i.e., a pair of source and target benchmark tasks from Table 2), the modeling pipeline retains the backbone representation trained on the source task and finetunes a task-specific head using data from the target task. This could help us answer questions such as: is LV segmentation a good pretext task for learning useful representations in patients with aortic stenosis? which architectures result in representations that generalize well across patient cohorts? How many echo studies do we need to annotate in order to finetune a pretrained model for a given downstream task? Answers to these questions can inform practical modeling choices and data collection requirements.

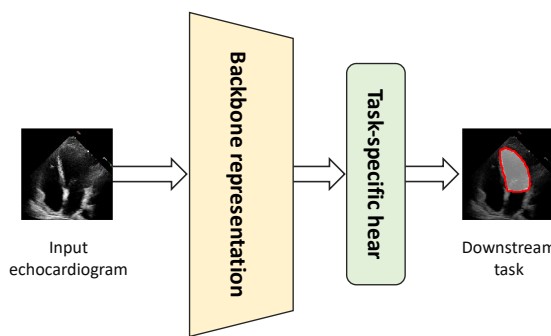

Figure 2: Depiction of a vision modeling pipeline $\mathcal{M}$.

## 3.2 The ETAB evaluation protocol

In many practical scenarios, we might be interested in evaluating how well a visual representation pretrained on an external data set captures echocardiographic features, e.g., private echocardiogram data, data for other cardiac imaging modalities, or even non-medical data sets (such as ImageNet). The echocardiographic task adaptation benchmark (ETAB) is a unified evaluation protocol that uses the core benchmark tasks in Table 2 to evaluate the usefulness of a given (pre-trained) visual representation for a wide variety of common downstream tasks in echocardiography.

Let $\mathcal{K} = \{1, \ldots, K\}$ be the benchmark categories, and let $\mathcal{T}_k = \{t_{1,k}, \ldots, t_{T_k,k}\}$ be the tasks within category $k \in \mathcal{K}$. (In Table 2, we have $K = 4$ task categories.) Let $\mathcal{D}_{t,k}$ be the data set associated with the $t$-th task within the $k$-th category. Let $\mathcal{R}_\theta$ be a pre-trained representation with an architectural specification $\mathcal{R}$ and parameters $\theta$. For each task $t_k$, we construct a model $\mathcal{M}_{\theta,\phi(t_k)} = \mathcal{R}_\theta \circ h_{\phi(t_k)}$, where $h_{\phi(t_k)}$ is a model head (e.g., a linear layer) that is specific to task $t_k$ with parameters $\phi(t_k)$. For each task $t_k$, the corresponding data set $\mathcal{D}_{t,k}^n = \{(X_{t,k}^i, Y_{t,k}^i)\}_{i=1}^n$ is used to optimize the parameters of the task-specific head $\phi^*$, with the representation parameters $\theta$ fixed. Let $\mathcal{E}_{t,k}$ be the evaluation metric used to assess performance on task $t_k$, which we assume takes on values in $[0, 1]$. The ETAB score of a pre-trained representation $\mathcal{R}_\theta$ is thus defined as:

$$\text{ETAB}_k^n(\mathcal{R}_\theta) \triangleq \mathbb{E}_{t \sim P_{\mathcal{T}_k}} \mathcal{E}_{t,k} \left[ \mathcal{M}_{\theta,\phi^*(t_k)}(\mathcal{D}_{t,k}^n) \right], \ \text{ETAB}^n(\mathcal{R}_\theta) \triangleq \mathbb{E}_{k \sim P_{\mathcal{K}}} \text{ETAB}_k^n(\mathcal{R}_\theta). \quad (1)$$

The ETAB score can be evaluated for different values of $n$ to assess the sample efficiency, i.e., transferability, of a pre-trained representation. We implement the metric in (1) by assigning equal weights to all benchmark tasks across all task categories, i.e., $P_{\mathcal{T}_k}$ and $P_{\mathcal{K}}$ are uniform distributions. A weighted ETAB score (with non-uniform choice of $P_{\mathcal{T}_k}$ and $P_{\mathcal{K}}$) can be defined to reflect the relative importance or frequency of the different echocardiographic tasks in Table 2 in real-world settings.

The ETAB protocol is model-agnostic, hence it can serve as a benchmark against which all existing and future architectural choices $\mathcal{R}$ and pre-training algorithms can be evaluated. This includes representations pre-trained using supervised, semi-supervised, self-supervised, and generative approaches. Unlike the individual benchmark tasks in Section 3.1—which assess specific downstream or transfer learning scenarios—the ETAB protocol assesses the general usefulness of a visual representation to

echocardiographic tasks through a single procedure and metric. Currently, the total number of benchmark tasks involved in ETAB is 31. We envision that the number of ETAB tasks will grow as more data sets and annotations for echocardiograms are released for public access over time.

To the best of our knowledge, ETAB is the first systematic benchmarking setup dedicated to evaluating echocardiographic representations. Our benchmarking setup can be considered as the domain-specific analogue of the general benchmark for vision tasks developed in [19, 20]. (Further discussion on related work is provided in the supplement.) Models that perform well on general benchmarks may not necessarily be as competitive on downstream tasks specific to echo data, which creates the need for benchmarks that are tailored to the tasks most relevant to cardiovascular problems. A schematic depiction of the protocol is provided in Figure 3.

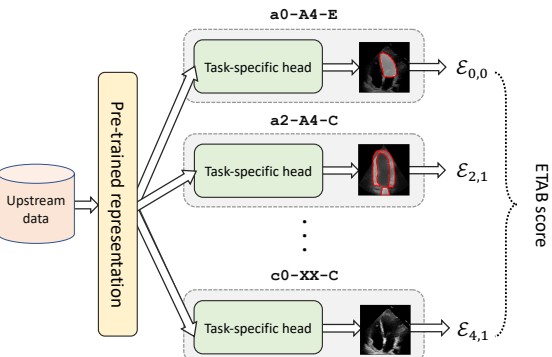

Figure 3: Schematic depiction of the ETAB protocol.

## 4 The ETAB Software Library

The ETAB library is a PyTorch-based implementation of our benchmarking framework that provides a unified and easy-to-use API for running benchmark experiments out-of-the-box, developing new vision modeling pipelines for echocardiography and evaluating pre-trained representations using the ETAB score. The ETAB library offers two key features:

- A unified API for loading and processing publicly-accessible echocardiography datasets.
- An easy-to-use API for creating modeling pipelines that supports a wide variety of backbone architectures and covers the benchmark tasks in Table 2.

The ETAB package can be installed via https://github.com/ahmedmalaa/ETAB. In this Section, we describe the key features of the package along with representative results for benchmark experiments.

### 4.1 Key features and examples of usage

**ETAB data sets.** The ETAB library supports a unified API for all public-access data sets listed in Table 1. The `ETAB_dataset` class is a container for echocardiography data with a multitude of options that specify the data source, echo view, annotations and parameters of the echo study clip (e.g., resolution and frame size, clip length, frame sampling rates). An example for instantiating a data set of apical 4-chamber echo studies with LV EF measurements is provided below:

```
import etab

echonet = ETAB_dataset(name="echonet", target="EF", view="A4CH",
                       video=False, normalize=True, frame_l=224,
                       frame_w=224, clip_l=16, fps=50, padding=None)
```

The `ETAB_dataset` comes with built-in functionalities for loading and processing the data. A detailed description of these functionalities can be found in the ETAB package documentation.

**ETAB model zoo.** The ETAB package supports a variety of built-in vision modeling pipelines. Currently, the package supports **12** backbone representations that belong to the convolutional neural network (CNN) and vision transformer (ViT) families. Note that some benchmark tasks (e.g., estimation of LV ejection fraction) are defined with respect to video clips rather than still images, whereas other tasks and datasets are limited to 2D images. In the current release of ETAB, we restrict the backbone representations to frame embeddings and use these representations repeatedly over sequences of images. By restricting the backbone representations to frame embeddings, we ensure that the backbone architecture can be used systematically across all benchmark tasks. In Section 4.3,

we describe the task-specific heads for all benchmark categories in Table 2. The ETAB model zoo (all of the supported backbone and model heads) is available in the Supplementary material.

**ETAB score computation.** To compute the ETAB score for a given pre-trained representation, the user can load the pre-trained weights for a PyTorch implementation of any of the supported backbones, and then invoke the `ETABscore` function to evaluate the quality of the representation as follows:

```
from etab.scores import ETABscore
from torchvision.models import resnet50

backbone = resnet50(weights="IMAGENET1K_V1")

etab_score = ETABscore(backbone_architecture="ResNet-50",
    backbone_model=backbone)
```

In the example above, we load a ResNet-50 model pre-trained on the ImageNet-1K data set and then evaluate its ETAB score, i.e., measure how well a ResNet-50 trained on non-medical imaging data captures echocardiographic features. By default, the ETAB score is evaluated assuming a uniform weight for all benchmark tasks and with the pre-trained weights being frozen across all tasks. The `ETABscore` function also supports full finetuning and weighted sampling of benchmark tasks.

**Evaluation metrics.** We use the following evaluation metrics within the ETAB evaluation protocol in Equation (1). For all segmentation tasks, the metric $\mathcal{E}_{t,k}$ corresponds to the DICE score. For classification tasks, $\mathcal{E}_{t,k}$ is defined as the area under the ROC curve (AUC-ROC). For regression tasks (i.e., predicting LV EF), we use $R^2$ as the metric $\mathcal{E}_{t,k}$. All metrics are computed on held-out samples. Note that, while we use different metrics for the different tasks, all metrics are defined in the normalized range $[0, 1]$ (higher is better). To compute the ETAB score for a given baseline, we run all benchmark tasks in Table 2 and average the resulting evaluation metrics.

## 4.2 ETAB modeling pipelines

As discussed in the previous Section, ETAB supports backbone representations $\mathcal{R}_\theta$ that belong to the CNN and ViT architectural families. The CNN backbones include: ResNet [22], ResNeXt [23], DenseNet [24], Inception [25], MobileNet [26] and ConvNext [27]. The ViT backbones include: Mix Transformer encoders (MiT) [28], Pyramid Vision Transformer (PVT) [29], Multi-scale vision Transformer (ResT) [30], PoolFormer [31], and UniFormer [32]. In what follows, we describe the task-specific heads $h_\phi$ for the four benchmark categories in Table 2.

(•) *Task-specific heads for cardiac structure identification benchmarks.*

The "cardiac structure identification" tasks (•) involve semantic segmentation of ventricular and other anatomic structures. We consider the following segmentation heads: U-Net [33], U-Net++ [34], MAnet [35], Linknet [36], PSPNet [37], DeepLabV3 [38], TopFormer and SegFormer [28]. Depending on the architecture of the selected backbone representation, the default segmentation head used in the `ETABscore` function is either a U-Net or a SegFormer.

(•) *Task-specific heads for cardiac function estimation benchmarks.*

This category of tasks involve estimating a patient's cardiac output from an echocardiography video. For all tasks in this category, we apply the backbone representations to obtain frame-level embeddings for the sequence of frames in each echo study. The task-specific heads are variants of recurrent neural networks (RNN) applied to these frame embeddings to obtain a representation for the sequence.

(•, •) *Task-specific heads for view classification and clinical predictions.*

For all benchmark tasks in this category, we considered baseline models that train attach a simple linear layer on top of the backbone representation to evaluate the output prediction.

The `ETABmodel` class in ETAB provides a unified sklearn-like API for all vision modeling pipelines. Examples of usage of the ETAB modeling pipelines are provided in the demo notebooks within the ETAB package documentation. Evaluations of the ETAB scores for state-of-the-art backbone models are maintained and regularly updated on the online ETAB leaderboard. In the rest of this Section, we provide sample results for applying the vision modeling pipelines to the benchmark tasks in Table 2.

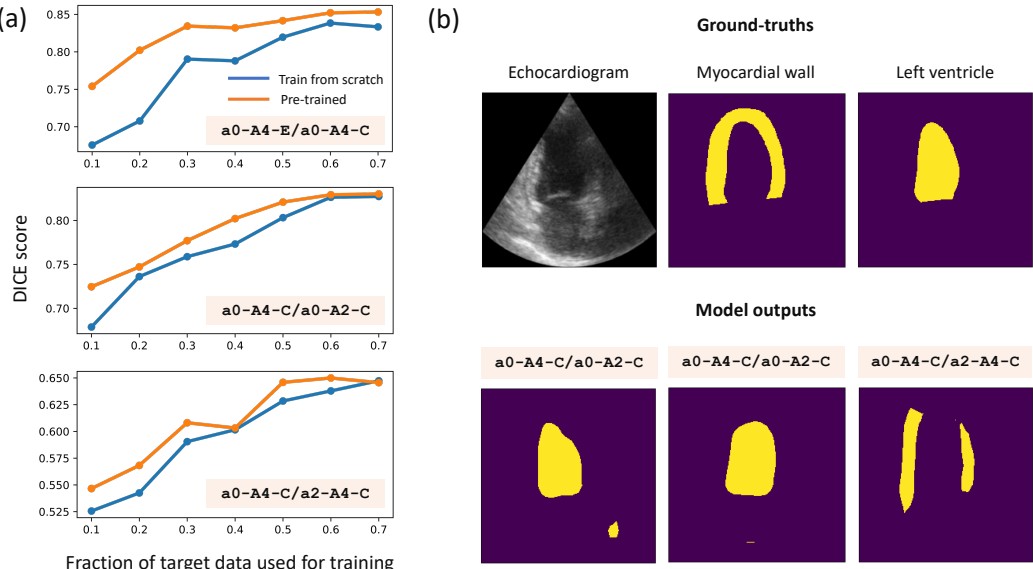

Figure 4: **Results for the cardiac structure identification benchmarks (●).** (a) Performance of the SegFormer model head with a ResNet-50 backbone with and without pre-training on the source data in each transfer learning benchmark. (b) Ground-truth traces (top) and samples of the recovered segmentation by the model (bottom).

## 4.3 Transfer learning benchmarks

In this Section, we consider the **cardiac structure identification benchmarks (●)** in Table 2. Here , the goal is to segment various anatomic structures of relevance to cardiac diagnostics. In particular, the tasks cover segmentation of the left ventricle (LV), the left atrium (LA), and myocardial wall (MY). LV segmentation is used to compute cardiac contractile function parameters (e.g., systolic/diastolic volumes, ejection fraction, and myocardium mass), which are key indicators of cardiac health [39]. Similarly, segmental evaluation of the MY and LA confer diagnostic value since MY wall thickness and LA enlargement are associated with pathological conditions such as hypertrophy [40].

Based on these benchmarks, we consider 3 experimental setups designed to test the transferability of features across data sets, echo views and tasks. Setup `a0-A4-C/a0-A2-C` can be used to assess the transferability of ML models across 4- and 2-chamber views, setups `a0-A4-C/a1-A4-C` and `a0-A4-C/a2-A4-C` assesses transferability across targets, whereas `a0-A4-E/a0-A4-C` tests transferability across data sets. Setups `a0-A4-E/a0-A2-C` and `a0-A4-E/a2-A2-C` are more challenging as they involve changing data sets, views and labels between the source and target tasks.

In Figure 4 (a), we show the performance of the SegFormer head with a ResNet-50 backbone for setups `a0-A4-E/a0-A4-C`, `a0-A4-C/a0-A2-C` and `a0-A4-C/a2-A4-C`. We evaluate the sample efficiency of the baseline model by fixing the number of training examples in the source task, and varying the number of training examples within the target task. This setup emulates the real-world scenario where we would have few echocardiographic studies extracted from hospital records for a cohort of interest that are manually annotated by a cardiologist, and we would like to adapt a model pre-trained on data from a different hospital to the cohort under study. Stronger baselines would enable us to collect and manually annotate less data points.

The results in Figure 4 (a) demonstrate the relative difficulty of the different transfer learning tasks in terms of the number of (target) examples required for competitive performance. For each of the benchmark tasks, we compare an adaptation pipeline that pre-trains the model on the source task with training from scratch on the target task. For the `a0-A4-E/a0-A4-C` and `a0-A4-C/a0-A2-C` setups, we observe that pre-training on source task significantly outperforms supervised learning from scratch when the number of target samples is as few as 16 target examples. From a practical perspective, this means that we can use these benchmarks to select adaptation pipelines for small-scale studies involving echocardiography data (e.g., clinical trial data), and expect the selected model to per-

| | Benchmark | U-Net | U-Net++ | MAnet | Linknet | PSPNet | DeepLabV3 |
|---|---|---|---|---|---|---|---|
| Cardiac struct. identification | ● a0−A4−C/a0−A2−C | 0.885 | **0.893** | 0.752 | **0.892** | 0.847 | 0.840 |
| | ● a0−A4−C/a1−A4−C | 0.729 | 0.705 | **0.739** | 0.481 | 0.684 | 0.659 |
| | ● a0−A4−C/a2−A4−C | 0.540 | 0.494 | 0.539 | **0.709** | 0.532 | 0.666 |
| | ● a0−A4−E/a0−A4−C | **0.909** | 0.804 | **0.898** | 0.896 | **0.900** | **0.909** |
| | ● a0−A4−E/a0−A2−C | **0.891** | 0.774 | 0.882 | **0.894** | 0.883 | 0.879 |
| | ● a0−A4−E/a2−A4−C | 0.634 | 0.603 | **0.691** | 0.580 | **0.707** | 0.671 |

Table 3: **Performance of different segmentation heads on cardiac structure identification benchmarks.** In all experiments, we use 32 training examples in the target task. All baselines use ResNet-50 backbone as the encoder architecture. Reported numbers are DICE scores on a test set; bold values correspond to best performing baseline.

form competitively on the data set at hand even if the number of samples is small. Samples of the boundaries of LV and MY detected by the adapted models are illustrated in Figure 4 (b).

Figure 4 (a) shows that, while transfering learned features across data sets and across echo views is feasible, transferring representations across different labels (i.e., LV and MY) is a difficult task. We observe that for setup `a0−A4−C/a2−A4−C`, pre-training on the source task was no better than training from scratch on target data. The difficulty of this benchmark task inspires future research that focuses on developing adaptation pipelines that use other approaches, such as self-supervised or multi-task learning, to improve the transferability of features across the LV and MY labels.

Table 3 demonstrates the performance of all segmentation heads with the ResNet-50 backbone on 6 transfer learning benchmarks. We observe that no modeling pipeline uniformly outperforms others on all benchmarks, which suggests that our benchmark suite can be used for model selection in new downstream tasks. Overall, we found that transfer learning benchmarks involving label shift (e.g., `a0−A4−C/a1−A4−C` and `a0−A4−C/a2−A4−C`) are more difficult than benchmarks involving shift in data sets or views. Benchmarks that involved label shifts also exhibited larger variance in the observed performance of the different modeling baselines.

| Representation | Weights | ETAB score | Representation | Weights | ETAB score |
|---|---|---|---|---|---|
| MobileNet-V2 | ImageNet-1K | 0.783 | PoolFormer-S24 | ImageNet-1K | 0.692 |
| ResNet-50 | Fully finetuned | 0.769 | MiT-B2 | Fully finetuned | 0.691 |
| MobileNet-V3-Large | Fully finetuned | 0.749 | ResNet-50 | ImageNet-1K | 0.689 |
| ResNet-18 | ImageNet-1K | 0.702 | MiT-B2 | ImageNet-1K | 0.653 |
| ResNet-34 | ImageNet-1K | 0.699 | ConvNext-Base | Fully finetuned | 0.647 |

Table 4: **ETAB scores for various backbone representations.**

## 4.4 Evaluating pre-trained representations using the ETAB protocol

Finally, we compare various pre-trained backbone representations with respect to their ETAB scores and report the results in Table 4. Here, we evaluate the ETAB scores by tuning the different backbone models to tasks `a0−A4−E`, `a0−A4−C`, `a0−A2−C`, `a1−A4−C`, and `a1−A2−C`, with equal weights for all tasks. We explore two variants of each backbone model: freezing the pre-trained ImageNet-1K weights and tuning the head, and fully finetuning the entire model for each benchmark task. The backbone models included CNN architectures (ResNet, MobileNet, ConvNext), as well ViT models (MiT and PoolFormer). Table 4 lists the ETAB scores for all backbone models. A visualization for the break down of the task-specific ETAB scores is available in the online leaderboard.

Overall, we found that the CNN-variants outperformed their ViT counterparts. Moreover, full finetuning on the ETAB data sets did not significantly improve the ETAB scores for most backbone models. An interesting use case for our the ETAB framework is to explore whether pre-training backbone representations on large-scale medical imaging data sets, such as RadImageNet [41], could outperform the ImageNet-pretrained backbone with respect to the ETAB score.

# 5 Conclusion

Deep learning can help automate the echocardiography workflow and refine our understanding of heart disease subtypes using cardiac ultrasound data. The scarcity and sporadicity of publicly accessible echocardiogram data sets create a need for standardized benchmarks in order to encourage engagement in research applying state-of-the-art representation learning methods to echocardiographic data. This paper takes a first step towards building a standardized framework for benchmarking the quality of representations for echocardiograms. We hope that the benchmark tasks and evaluation methods presented in this paper will lower the barrier to translating advances in visual representation learning into impactful applications in the medical domain.

## Acknowledgments and Disclosure of Funding

Funding to support this research was provided for by the Eric and Wendy Schmidt Center at the Broad Institute of MIT and Harvard (https://www.broadinstitute.org/ewsc.).

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
