# Supplementary Material

**Statement on ethical implications.** The ETAB framework provides a standardized API for loading and processing all datasets for the sake of model development and training, but it does not alter, distribute or directly share the datasets. Please make sure to follow the terms of the respective Research Use Agreements for all datasets upon access. Links to the data licenses for all datasets are available in the ETAB documentation. Please note that the ETAB library is **not meant to be used within clinical workflows**. Please use ETAB for research and model derivation purposes. Please also note that any sources of bias inherent in the datasets covered by ETAB might be reflected in the derived modeling pipelines and the corresponding ETAB scores.

**Computing resources.** All experiments in this paper were conducted using computational resources on Google Cloud Platform (GCP). Each individual benchmark experiment was conducted using an NVIDIA A100 GPU on a GCP virtual machine.

**Dataset details.** Table S1 provides an overview of the features and characteristics of all datasets currently incorporated into the ETAB benchmarking framework.

|  | **EchoNet** | **CAMUS** | **TMED** |
|---|---|---|---|
| Sample size | 10,036 | 500 | 2,341 |
| Echo views | AP4CH | AP4CH, AP2CH | PLAX, PSAX and other |
| Annotations | LV traces and LVEF | LVEF, ES/ED frames, LV epi-, endo-cardium and LA segments | AS diagnoses, view labels |
| Labeled data | All studies | All studies | 260 studies |
| Frame sizes | 112x112 pixels | Variable per study | 64x64 pixels |
| Video clips | Available | Available | Unavailable |

Table S1: **Details of all echocardiographic datasets in ETAB.**

# A    Related work

**Deep Learning for echocardiography.** Previous work on applying deep learning models to echocardiography has focused on automating the echo pipeline, i.e., training a model to conduct the cardiac measurements and traces that would otherwise be normally performed manually by clinicians. Previous work on the first category of tasks (● Cardiac structure identification) developed various approaches for segmenting cardiac structures based on single frames or sequences of frames in an echo clip. Existing models have either relied on completely data-driven approaches [34, 39] or a combination of deep learning modeling and statistical shape models [42]. Deep learning models for segmenting the left ventricle and the myocardial shape included U-Nets, LSTMs and 3D CNNs and DeepLabV3 [7, 43, 44]. Our benchmarking framework allows testing these different methodologies in a systematic fashion. The different data structures in the EchoNet, CAMUS and TMED data sets allow benchmarking the tasks of LV and myocardial segmentation using both static and temporal data. Previous work on (● Cardiac function estimation) developed models for predicting LVEF on the basis of both 2D images or 3D video frames using 3DCNNs or recurrent neural network architectures (RNN) [7, 14, 45]. Our benchmarks cover both setups; it can enable evaluating LV segmentation based on 2D images by replacing the LSTM prediction head with a repeatedly applied feedforward neural network. Our benchmarking framework can also be used to evaluate the adaptability of existing models for (● view recognition) , e.g., [46], on new views (different from the ones originally involved in model training) as our benchmarks include apical 2- and 4-chamber views along with the parasternal views.

**Benchmarks for computer vision models.** To the best of our knowledge, ETAB is the first systematic benchmarking setup dedicated to evaluating echocardiographic representations. Our benchmark can be considered as the domain-specific analogue of the general benchmark for vision tasks developed in [19, 20]. Various benchmarking frameworks for image models has been developed; in addition to VTAB, other examples include the Visual Decathlon [47]and Meta-Dataset [48], etc. While general image benchmarks are useful for evaluating the quality of general visual representations; domain-specific benchmarks better reflect the adaptability of visual representations to tasks most relevant to

cardiac ultrasound, i.e., models that perform well on general benchmarks may not necessarily be the best models to finetune for tasks involved in echocardiography.

| Available backbones | | Reference |
|---|---|---|
| *Convolutional Neural Networks (CNN)* | ResNet
ResNeXt
DenseNet
Inception
MobileNet
ConvNext | [22]
[23]
[24]
[25]
[26]
[27] |
| *Vision Transformers (ViT)* | Mix Transformer encoders (MiT)
Pyramid Vision Transformer (PVT)
Multi-scale vision Transformer (ResT)
PoolFormer
UniFormer
Dual Attention Vision Transformers (DaViT) | [28]
[29]
[30]
[31]
[32]
[49] |

Table S2: **List of all available backbones in ETAB.**

| Available task-specific heads | |
|---|---|
| Classification and regression heads (still image) | Standard linear probe |
| Classification and regression heads (video clips) | RNN + Linear output layer
LSTM + Linear output layer |
| Segmentation heads | U-Net
U-Net++
MAnet
Linknet
PSPNet
DeepLabV3
SegFormer
TopFormer |

Table S3: **List of all task-specific heads in ETAB.**

| Representation | ETAB scores | | | |
|---|---|---|---|---|
| | 🔴 | 🔵 | 🟢 | 🟡 |
| ResNet50-ImageNet-S | 0.85 | 0.61 | 0.68 | 0.55 |
| ResNet50-ImageNet-SSL | 0.82 | 0.59 | 0.61 | 0.53 |
| ResNet50-3DSeg8-S | 0.89 | 0.73 | 0.72 | 0.63 |

Table S4: **Breakdown of the ETAB scores by task categories for various representations.**

## B    Description of the benchmark task codes

Each task adaptation benchmark is encoded through an 5-character code that describes the properties of the benchmark tasks of the form XX-XX-X. The characters within each code encode the tasks as:

- **Characters 0-1:** Task identified

- **Characters 2-3:** View identifier
- **Characters 4-5:** Dataset identifier

The data sets are encoded as follows **E**: EchoNet, **C**: CAMUS, and **T**: TMED. The views are encoded as follows: **A2**: Apical 2-chamber, **A4**: Apical 4-chamber, **PL**: parasternal long axis, and **PS**: parasternal short axis. The 4 tasks categories are encoded as **a**, **b**, **c** and **d**. The task identifiers are defined as:

**(a) Cardiac Structure Identification Tasks**

- 0: Segmenting the left ventricle (LV)
- 1: Segmenting the left atrium (LA)
- 2: Segmenting the myocardial wall (MY)

**(b) Cardiac Function Estimation Tasks**

- 0: Estimating LV ejection fraction
- 1: Classifying end-systole and end-diastole frames

**(c) View Recognition Tasks**

- 0: Classifying apical 2- and 4-chamber views
- 1: Classifying parasternal short and long axis views

**(d) Clinical Prediction Tasks**

- 0: Diagnose cardiomyopathy
- 1: Diagnose aortic stenosis

## C    Experimental setup

The ETAB software library can be accessed through this link: https://github.com/ahmedmalaa/ETAB. The library provides thorough documentation and step-by-step instructions for the ETAB benchmarking framework, as well as code snapshots to reproduce our experiments. The documentation provides a description of the ETAB API which can be used to run any benchmark experiment out-of-the-box. The ETAB website also contains a continuously updated leaderboard with a list of all supported backbone representations along with their corresponding ETAB scores.

**Baselines and hyper-parameters.** The ETAB library currently supports 12 backbone representations. We define a backbone representation as a general-purpose frame embedding for echocardiograms that is independent of the benchmark task category, whereas the head changes based on the task. The backbone representations supported in ETAB fall into two categories: convolutional neural networks and vision transformers. The list of all backbone representations in ETAB are listed in Table S2. Links to the hyperparameter configurations and architectural specifications for all backbones are provided in the online ETAB leaderboard.

All experiments were conducted using an SGD optimizer with a batch size of 32, a learning rate of 0.001 and 50 training epochs. For each benchmark task, the corresponding echocardiogram dataset was split into a 60% training sample, 10% validation sample and a 40% testing sample.

**Breakdown of the ETAB benchmark results.** The ETAB protocol allows evaluating pre-trained representations both on an aggregate level or with respect to each task category. Table S4 shows the performance of different pre-trained representations with respect to the 4 task categories.