# OpenReview forum: "ETAB: A Benchmark Suite for Visual Representation Learning in Echocardiography"
_NeurIPS.cc/2022/Track/Datasets_and_Benchmarks — NeurIPS 2022 Datasets and Benchmarks _

### Official Review · Reviewer_sq9k · 2022-07-21
**A promising medical imaging benchmark, but needs additional work**

**Rating:** 7
**Confidence:** 4

**Strengths:**

- The benchmark is very promising. Moving beyond traditional benchmark datasets, which are often narrowly constrained to iid assumptions by task and dataset, is critical for practical development of applied medical ML.
- Incorporating small training sizes (16+ training examples) is a nice focus area, as few-shot medical benchmarks is a critical research area.
- The downstream tasks cover a nice range of realistic use cases (segmentation, cardiac function estimation, view classification, clinical attributes).
- A metadata with a formal evaluation protocol across datasets would be a nice contribution

**Weaknesses:**

The most significant weaknesses of this work are
- A lack of clarity outlining the structure of the overall experimental pipeline in terms of measurable sub-components.
- Reported results are terse or in some cases missing (i.e. all non-ResNet50 performance numbers). Currently the manuscript lacks a full  breakdown of results even for the ResNet50 backbone use case that forms most of the paper's results. This incomplete reporting makes it difficult to follow the overall merits of the benchmark.

Clarity of results could be substantially improved. The paper proposes an evaluation strategy that covers a large number of model configurations, spanning architectures, datasets, pretraining strategies, and downstream task performance inclusive of some adaptation/transfer approaches. This is a very large cross product. While the gestalt ETAB score provides some useful assessment of representation learning quality (to the points raised in lines 146-147 around how individual scores assess specific transfer learning scenarios) a complete reporting of the individual source/target scores would make this paper much easier to follow.

Figures 3 and 4 include subsets of tasks. Why not report performance for all task categories?  In general, I feel the paper is missing reporting for many intermediate results (e.g., performance metrics for the source models) as well as not providing a consistent report for all downstream tasks.

The paper only includes results for ResNet50 architectures. Lines 173-176 states that other backbone architectures were evaluated, including vision transformers (ViT-L/16, Swin-ViT) and multi-layer perceptrons (MLP-mixer), however these performed worse than the ResNet50. While these results are stated as included in the supplementary material (line 176) they are actually not included. The supplement (which is quite short) includes pointers to the ETAB GitHub, but many links are broken and no results or leaderboard are available. I feel this is a substantial limitation and the paper would be substantially improved if we could observe ETAB scores across more than 1 family of backbone architectures.  I also find that not including the performance for the transformer and MLP architectures due to inferior performance to the ResNet50 is counter to the utility of a leaderboard benchmark, where these backbones are very reasonable baselines and increasing in popularity.

Some of the datasets may have availability issues. While EchoNet has a complete DUA process and provides archival information the other 2 datasets seem problematic. The "Tufts Medical Echocardiogram Dataset" is currently a dead link. CAMUS is from a challenge dataset and it's unclear from the website if the data is available to researchers who are not participating in the challenge (which ends in 31 Dec 2024). I was unable to fine an explicit discussion of licensing terms for CAMUS. Sometimes medical challenge datasets are only available for a given challenge. What guarantees do you have that these datasets will be available to researchers moving forward?

**Additional Feedback:**

- There is no author statement bearing all responsibility in case of violation of rights, etc., and confirmation of the data license.

- Given the experiments run, the manuscript should include an answer to section 3(d) in the checklist on the compute required to run experiments. Feels like the answer should be [Yes] (or at least [No] + justification) instead of [N/A]

- Given the scale of n in Figure 3a x-axis (n=16 to 256) it would be clearer to just use the original integer vs. the log_10 transform and label the axis something more intuitive like "n training examples".

**Clarity:**

While compact for bucketing results or various model configurations for reporting results, I found the benchmark task coding (e.g., EA40 CA22) very frustrating to parse while reading. This seems more useful as syntax for submitting to a leaderboard vs. something that enables clearer discussion for highlighting results in a manuscript.

I think the benchwork would benefit considerably from anchoring on a single attribute across source and target (same/different task, same/different pretraining dataset) and discussing the performance differences.

Given the mix and heterogeneity of the datasets, Table 1 would benefit from more summary information (or at least a more complete table in the supplement). Currently it's very hard to get a gloss of these 3 datasets in their "off-the-shelf" state. For example, what are the resolution of the images? What is a normalized unit of samples, such as number of total frames, that can be used to compare their relative scales? What are the counts of unlabeled vs. labeled across task categories (segmentation, view classification, severity ratings). Some of these details are found in Table 2, but key properties, like TMED not containing any cardiac function or structure tasks, containing only still images, etc. would be much clearer if Table 1 had clearer attribute columns.

Figure 3 was confusing. Subfigure A contains results for 3 benchmark segmentation models/task . Why these three and not all models? Why are the 3 benchmarks (EA40CA40, CA40CA20, CA40CA42) for A different than (CA40CA20, EA40CA20, EA40CA22) used for B? Subfigure B contains gold standard (top) and U-Net predictions (bottom), but what does gold standard mean for the echocaediogram image view? If the point is to show the deltas between gold and predicted segmentation masks, there are clearer plots you could use here.

**Correctness:**

- Item 3(c) in the checklist asks for error bars with respect to running with multiple seeds, but none of the reported experiment results report any measure of standard error or standard deviation.

- There do not appear to be any obvious errors in work.

**Documentation:**

The current GitHub repository contains minimal documentation, with most files containing only placeholder text. It is not currently possible, for example, to replicate or run any of the benchmark tasks out-of-box.

**Ethics:**

There is no real discussion of ethical implications in this paper beyond mention of the ethics review guideline in the checklist.
While this benchmark reuses existing datasets (and therefore inherits existing patient protections of the dataset) I feel there should be some discussion of this, at least justifying why there are no significant new ethical concerns introduced by creating this benchmark.



**Relation To Prior Work:**

This work builds on 3 existing echocardiographic datasets by proposing the ETAB benchmarking suite.

A related work section is provided in the supplemental materials that provides a high-level overview of the state of deep learning for echocardiography.


**Summary And Contributions:**

The work presents a benchmark suite of tasks for evaluating the performance of learned visual representations for use in echocardiography. This benchmark, the echocardiographic task adaptation benchmark (ETAB), provides a meta-dataset constructed from 3 existing echocardiogam research datasets. ETAB formalizes several categories of source and target task categories, and defines an aggregate ETAB score for estimating performance across a these benchmark categories.

This benchmark enables evaluating a number of backbones and other representation learning approaches across a heterogenous mix of echocardiographic tasks and datasets, which is critical for assessing robustness and other properties of representation learning methods.

---

> ### Author Response · Authors · 2022-08-29
> **Thanks for your thoughtful comments + Updates and changes in ETAB**
>
> We would like to thank the reviewer for the constructive and thoughtful comments which greatly helped us improve the quality of our benchmark framework. We have used the rebuttal period to incorporate major changes to the software and the submission to address all your comments---a detailed point-by-point description of these changes along with a response to your comments are provided below. Please also refer to the combined response above for an overall summary of the improvements and changes that have been incorporated in the [ETAB software library](https://github.com/ahmedmalaa/ETAB) to address your comments.
>
> &nbsp;
>
> **Improving clarity of the experimental pipeline**
>
> &nbsp;
>
> *ETAB score breakdown and complete reporting of task-specific performance*
>
> Please note that the breakdown of the ETAB score by task category was already reported in Table S1 in the supplementary material of the original manuscript. To address your comment, the [online ETAB leaderboard]( https://github.com/ahmedmalaa/ETAB/blob/main/docs/leaderboard.md) now allows visualizing a performance breakdown in terms of the task-specific scores for each backbone representation.
>
> In Figure 3, the goal was to assess the few-shot capabilities of representations transferred across source and target tasks that differ only in one attribute (the data source, the view, or the annotation), with the other attributes fixed. (This experiment resonates with a suggestion you bring up later in your review.) The reason for selecting these 3 benchmark segmentation tasks (EA40 -> CA40, CA40 -> CA20 and CA40 -> CA42) is that the EchoNet and CAMUS dataset have the same views and annotations, which enables running a controlled experiment with one attribute changing at a time (i.e., In EA40 -> CA40 only the dataset changes, in CA40 -> CA20 only the view changes, and in CA40 -> CA42 only the annotation changes). Please refer to lines 208-216 in the original manuscript for a detailed discussion and justification of these choices.
>
> In the original manuscript, we visualized the results for the a select subset of the most insightful adaptation experiments (Figure 4)---visualizing the results for all possible pairs of the 19 benchmark tasks in ETAB is challenging due to the large number of adaptation scenarios. To enable consistent reporting of all possible adaptation setup, we are working on a leaderboard for adaptation experiments that is analogous to the [online ETAB leaderboard](https://github.com/ahmedmalaa/ETAB/blob/main/docs/leaderboard.md#etab-leaderboard) with drop-down menus to select the source and target tasks and an additional slider that allows tuning the number of target samples. In the interface for the currently operating [leaderboard]( https://github.com/ahmedmalaa/ETAB/blob/main/docs/leaderboard.md) for ETAB scores, we added an option that allows visualizing the exact breakdown of the task-specific performance. We hope that these modifications will address your concerns regarding the clarity and consistency of experimental reporting.
>
> &nbsp;
>
> *Clarity of benchmark task encoding*
>
> As you pointed out, the task coding was meant to create a unified language for describing a benchmark task for submitting to the ETAB leaderboard and specifying a benchmark experiment within the ETAB API and CLI. In the revised manuscript, we have modified the task coding to improve readability. For details on the new benchmark task encoding, please refer to [Section 2 in the ETAB documentation]( https://github.com/ahmedmalaa/ETAB/blob/main/docs/benchmark_tasks.md#benchmark-task-categorization-and-encoding).
>
> In a nutshell, the modified encoding improves readability by changing the ordering of the characters within the code to a more natural ordering that emphasize the task type and echo view over the data source, and introducing hyphens to separate the task, view, and dataset identifiers. Moreover, in the revised manuscript, we decreased the reliance on the codes to describe the benchmark tasks to improve readability.
>
> &nbsp;
>
> *Anchoring a single attribute across source and target*
>
> We agree with you that anchoring on a single attribute across source and target (same/different task, same/different pre-training dataset, same/different view) is a good experimental setup for testing feature transferability across these attributes. Please note that we have already done that in Figure 3(a). In this experiment, we show the performance of the U-Net segmentation baseline (with a ResNet-50 backbone) for tasks EA40 -> CA40, CA40 -> CA20 and CA40 -> CA42 (based on the old task encoding scheme). The three selected tasks are representative of transfer learning across data sets, views, and targets (labels), respectively. Please refer to lines 208-216 in the original manuscript for a discussion of this experiment. Also please note that such anchoring experiments cannot be conducted across all tasks as not all datasets share the same views or annotations.

---

> > ### Author Response · Authors · 2022-08-29
> > **Thanks for your thoughtful comments + Updates and changes in ETAB (Part 2)**
> >
> > &nbsp;
> >
> > *More detailed description of the datasets*
> >
> > We have added more detailed description of the 3 datasets in Table S1 in the Supplementary material of the revised manuscript. For each dataset, we specify the total number of frames, frame size and image resolution, available annotations, labeled vs. unlabeled data, etc.
> >
> > &nbsp;
> >
> > **More comprehensive baselines**
> >
> > Thank you so much for suggesting an expansion of the family of backbones under consideration. The updated ETAB software now supports the following:
> >
> > - An expanded set of 12 backbones that belong to both the convolutional neural net (CNN) and vision transformer (ViT) families. A complete list of backbone representations and task-specific heads can now be found in the [ETAB model zoo](https://github.com/ahmedmalaa/ETAB/blob/main/docs/benchmark_tasks.md#etab-model-zoo).
> > - A unified API for creating modeling pipelines using CNNs and ViTs.
> > - A leaderboard that keeps track of the best performing backbones based on the overall ETAB score as well as task-specific benchmarks (See [Section 4]( https://github.com/ahmedmalaa/ETAB/blob/main/docs/leaderboard.md) of the ETAB documentation).
> >
> > The future releases of ETAB will continue to expand the available modeling pipelines by incorporating more backbone architectures and pre-trained weights.
> >
> > &nbsp;
> >
> > **Availability of the datasets**
> >
> > We have double-checked that the TMED dataset link is currently active. Please visit: [https://tmed.cs.tufts.edu/](https://tmed.cs.tufts.edu/).
> >
> > We sympathize with your concerns on the future availability of the datasets under study. Our view of the ETAB framework is that it is an evolving project that will be updated to encapsulate all the available echocardiography datasets the community has releases over time. For instance, we are currently working on incorporating the Unity dataset into the next release for ETAB, and this new dataset will enable crafting new benchmark tasks that cover longitudinal strain analysis and refined LV measurements (please refer to the [benchmark suite]( https://github.com/ahmedmalaa/ETAB/blob/main/docs/benchmark_tasks.md#benchmark-task-categorization-and-encoding) in the documentation). We anticipate that more public datasets of echocardiograms will be released in the future. However, since we do not have control over future accessibility of all datasets, our goal is to provide a unified API for a meta-dataset encapsulating all public data available at any moment of time---we believe that this is the best one can do to help the community take full advantage of available public data for model development and testing.
> >
> > &nbsp;
> >
> > **Documentation**
> >
> > A first version of the software documentation for the ETAB benchmarking framework is now [available online](https://github.com/ahmedmalaa/ETAB#documentation). The documentation includes step-by-step instructions on how to install our framework, download and process datasets, and run any benchmark experiment out-of-the-box.
> >
> > The current version of the documentation is organized in 5 Sections:
> >
> > &nbsp;
> >
> > [Section 1: Datasets, Accessibility and Data Processing Tools](https://github.com/ahmedmalaa/ETAB/blob/main/docs/data_access.md)
> >
> > * [Overview of the Supported Echocardiography Datasets](https://github.com/ahmedmalaa/ETAB/blob/main/docs/data_access.md#Datasets)
> > * [Instructions for Dataset Access](https://github.com/ahmedmalaa/ETAB/blob/main/docs/data_access.md#Instructions-for-dataset-access)
> > * [Data Loaders and Processing Tools](https://github.com/ahmedmalaa/ETAB/blob/main/docs/data_access.md#data-loaders-and-processing-tools-demo-notebook)
> >
> > [Section 2: ETAB Benchmark Suite and Model Zoo](https://github.com/ahmedmalaa/ETAB/blob/main/docs/benchmark_tasks.md)
> >
> > * [Benchmark Task Categorization and Encoding](https://github.com/ahmedmalaa/ETAB/blob/main/docs/benchmark_tasks.md#benchmark-task-categorization-and-encoding)
> > * [ETAB Model Zoo](https://github.com/ahmedmalaa/ETAB/blob/main/docs/benchmark_tasks.md#etab-model-zoo)
> > * [Running a Benchmark Experiment Out-of-the-Box](https://github.com/ahmedmalaa/ETAB/blob/main/docs/benchmark_tasks.md#running-a-benchmark-experiment-out-of-the-box-demo-notebook)
> >
> > [Section 3: The ETAB Evaluation Protocol](https://github.com/ahmedmalaa/ETAB/blob/main/docs/etab_protocol.md)
> >
> > * [Description of the Evaluation Protocol](https://github.com/ahmedmalaa/ETAB/blob/main/docs/etab_protocol.md#Description-of-the-Evaluation-Protocol)
> > * [Computing the ETAB Score](https://github.com/ahmedmalaa/ETAB/blob/main/docs/etab_protocol.md#computing-the-etab-score)
> >
> > [Section 4: Leaderboard and Benchmark Results](https://github.com/ahmedmalaa/ETAB/blob/main/docs/leaderboard.md)
> >
> > * [ETAB Leaderboard](https://github.com/ahmedmalaa/ETAB/blob/main/docs/leaderboard.md#etab-leaderboard)
> > * [How to contribute?](https://github.com/ahmedmalaa/ETAB/blob/main/docs/leaderboard.md#how-to-contribute)
> >
> > [Section 5: Reproducibility Checklist](https://github.com/ahmedmalaa/ETAB/blob/main/docs/reproducibility_checklist.md)

---

> > > ### Author Response · Authors · 2022-08-29
> > > **Thanks for your thoughtful comments + Updates and changes in ETAB (Part 3)**
> > >
> > > In addition to the documentation, demo notebooks are provided at the end of each Section to illustrate basic use cases of ETAB to users of the library. You can explore these demo notebooks through this [link]( https://github.com/ahmedmalaa/ETAB/tree/main/notebooks).
> > >
> > > &nbsp;
> > >
> > > **Running a benchmark experiment out-of-the-box**
> > >
> > > We have worked on standardizing the API for benchmark task and model instantiation within the ETAB library to enable a simple an intuitive way for users to run any benchmark experiment out-of-the-box. A thorough documentation for the ETAB API is available to users through this [link]( https://github.com/ahmedmalaa/ETAB/blob/main/docs/benchmark_tasks.md#running-a-benchmark-experiment-out-of-the-box-demo-notebook), and a demo notebook for how to run a benchmark experiment is available through this [link](https://github.com/ahmedmalaa/ETAB/blob/main/notebooks/Demo%202%20-%20ETAB%20Benchmark%20Tasks.ipynb).
> > >
> > > &nbsp;
> > >
> > > **Additional comments**
> > >
> > > *Ethical concerns:* Please refer to the first Subsection in the revised Supplementary material for a statement and discussion on potential ethical concerns.
> > >
> > > *Data license:* In the documentation for ETAB, we acknowledge the data licenses for the datasets involved in ETAB. Please refer to [Section 2](https://github.com/ahmedmalaa/ETAB/blob/main/docs/data_access.md) of the ETAB documentation for more details.
> > >
> > > *Compute requirements:* All our experiments were conducted using a single A100 GPU. We have checked “Yes” for section 3(d) of the checklist, and added a description of the compute resources used to run our experiments in first paragraph of the Supplementary material in the revised version of the manuscript.

---

> ### Comment · Reviewer_sq9k · 2022-09-03
> **Nice updates to the benchmark repo**
>
> I thank the authors for their substantive improves to the software. The quality of the updated framework is very nice, with great documentation and tutorials.
>
> That said, feel the presentation and clarity of the manuscript can still be improved. For example,  The mapping syntax (eg “a0-A4-C/a0-A2-C") requires referencing the supplement to parse. This needs to be part of the main manuscript, as noted by another reviewer. This reflects a general trend where some material is present in the suppliment, but should be moved to the main text. The rebuttal period affords the authors 1 extra page to improve the manuscript's clarity by expanding or moving content, but the authors have not done so.
>
> The paper would be also be stronger if there was a more systematic analysis of ETAB scores across backbones. The numbers are present on the leaderboard but not discussed in the main paper.
>
> I have updated my score based on the strength of the benchmarking API itself and the belief in the merits of the application setting, but I hope the authors iterate on the manuscript to include these details .

---

> > ### Author Response · Authors · 2022-09-03
> > **Thank you for the follow up comments!**
> >
> > As we mentioned in our original response, your comments played a major role in improving our initial submission, and we thank you for posting follow up comments to further improve our paper. We agree with all your follow up comments---we did not address these in the revised paper due to limited time and space, but will make sure to address all in the final version of the paper.

---

### Official Review · Reviewer_Htsc · 2022-07-25
**Well written paper.**

**Rating:** 8
**Confidence:** 3
**Clarity:** It is well written for clinicians and…

**Strengths:**

Novelty. Well written. Strong main experiment and sub-experiment.

**Weaknesses:**

The GitHub website is not as complete as I would like it to be. However, I can understand that it is constantly being updated.

**Additional Feedback:**

Thank you for your amazing work.

**Correctness:**

I am not a computer scientist. From a clinician-scientist point of view, I believe that the method and design are very plausible.

**Documentation:**

Yes, the benchmark is publicly available and it can be used by anyone. It can support reproducibility.

**Ethics:**

There is not any ethical concerns.

**Relation To Prior Work:**

Yes it does.

**Summary And Contributions:**

This is a very well written paper and I believe that this benchmarking setup can later be used for other medical specialties. The use of AI in medicine requires a higher threshold of security and testing than other use cases because the difference between right and wrong output can mean life or death. The authors of this paper are suggesting a novel standardized approach to solving this problem.

---

> ### Author Response · Authors · 2022-08-29
> **Thank you for the feedback! + Updates**
>
> Thank you so much for your feedback! Please refer to the combined response above for a summary of the updates incorporated in ETAB during the rebuttal period.

---

### Official Review · Reviewer_ofDL · 2022-07-28
**Review for ETAB**

**Rating:** 5
**Confidence:** 4

**Strengths:**

This paper shows powerful usage in specific clinical tasks like Cardiac struct. identification, estimation, and clinical predictions. Translating the clinical evaluation protocols into ML-friendly tasks will bridge the gap in deep learning applications in the medical domain.


**Weaknesses:**

- **Novelty**: as addressed in the paper, "the domain-specific analogue of the general benchmark for vision tasks developed in [19,20]". Though the ETAB definition involving the formula is slightly different, it does not increase superiority in methodology.

- **Methods are not comprehensive**: benchmarking a dataset should consider SOTA methodology in model architecture and pretraining. Though the paper mentioned ViT and Swin-ViT in line 173, it's not included in the main context and supplementary. In the baseline for the cardiac function estimation task, it only mentioned LSTM to handle the video frames without justification. The other methods for video data like ViViT should at least be mentioned. Other methods like semi-supervised learning and contrastive learning, which achieved a good performance recently, should also be considered as a good pretraining strategy in addition to transfer learning.

- **Pipeline not user-friendly**: Github repo does not provide a user-friendly interface to reproduce the result. All the links at the bottem are invalid.

**Additional Feedback:**

Can you try to put the related work in the main context?

**Clarity:**

Including formula (1) does more harm than good. Evaluation metrics in Sec 4.1 is more straightforward than that formula.

Figure 3 should indicate what's meaning of x axis.



**Correctness:**

The claims appear to be correct, and the experiments are conducted in a sound way. The experimental design and evaluation are fine.

**Documentation:**

Need more guidance to reproduce their main results.

**Ethics:**

No ethical concerns.

**Relation To Prior Work:**

As addressed in the related work, it's a domain-specific analogue to [19, 20].

**Summary And Contributions:**

This paper provides an echocardiographic task adaptation benchmark for the various clinically-relevant task using publicly available datasets. Their benchmark evaluates some model architectures, pretraining techniques to standardize the evaluation method for clinical usage.

---

> ### Author Response · Authors · 2022-08-29
> **Thank you for your thoughtful feedback + responses and updates in ETAB**
>
> We thank the reviewer for the thoughtful comments. We have worked on addressing all your comments regarding the comprehensiveness of the modeling pipelines, and the incorporation of a user-friendly API in the ETAB library. Please refer to the combined response above for a general overview of the improvements and changes that have been incorporated in the ETAB software library to address these issues.
>
> &nbsp;
>
> **Novelty**
>
> We believe that ETAB contributes meaningfully to research in medical imaging by providing a novel software tailored to analyzing echocardiography data---to the best of our knowledge, no such tool has been developed before. The novel aspects of ETAB are in line with the scope of the Datasets and Benchmarks, these novel aspects are:
>
> - A unified API for loading and processing various publicly-accessible echocardiography datasets.
> - A comprehensive suite of (transfer learning and downstream) benchmark tasks tailored to echocardiography.
> - An easy-to-use API for creating modeling pipelines that supports a wide variety of backbone architectures and covers different types of downstream tasks of interest.
> - A unified evaluation protocol for pre-trained echocardiographic representations.
>
>  &nbsp;
>
> **Comprehensiveness of modeling pipelines**
>
> Thank you for suggesting an expansion of the family of backbones in ETAB to support a more comprehensive modeling pipeline. The updated ETAB software now supports the following an expanded set of 12 backbones that spans a wide variety of convolutional neural net (CNN) and vision transformer (ViT) architecture. A complete list of backbone representations and task-specific heads can now be found in the [ETAB model zoo](https://github.com/ahmedmalaa/ETAB/blob/main/docs/benchmark_tasks.md#etab-model-zoo) (in Section 2 of the ETAB documentation). Please note that ETAB is an evolving library; future releases will continue to expand the available modeling pipelines by incorporating more SOTA backbone architectures and pre-trained weights. Results from the newly supported modeling pipelines are incorporated into the [online ETAB leaderboard](https://github.com/ahmedmalaa/ETAB/blob/main/docs/leaderboard.md#etab-leaderboard).
>
> Regarding the handling of video data and the rationale behind using RNN-type heads, please refer to the [“video data vs still images”](https://github.com/ahmedmalaa/ETAB/blob/main/docs/benchmark_tasks.md#video-data-vs-still-images) in Section 2 of the ETAB documentation. In the current release of ETAB, we restrict the backbone representations to frame embeddings and use these representations repeatedly over sequences of images and defer the modeling of the temporal correlations between these embeddings to the head through variants of RNNs. The rationale for our focus on frame embeddings is that the ETAB score is designed to evaluate a pre-trained embedding over several diverse benchmark tasks. Because not all benchmark tasks are defined on video data, we need to disentangle the backbone architecture from the sequence modeling component of the modeling pipeline to enable a uniform evaluation protocol across all tasks.
>
> &nbsp;
>
> **User-friendliness of the ETAB library**
>
> In response to your comment, major modifications has been made to the ETAB software library to support a user-friendly API that enables users to easily load datasets, create modeling pipelines, running benchmark experiments and computing the ETAB score for general backbone representations in a succinct and modular manner. Please refer to the [ETAB documentation]( https://github.com/ahmedmalaa/ETAB#documentation) for a detailed showcase of the new features incorporated into the ETAB library.
>
> In addition to the thorough documentation, we provide a number of demo notebooks that include step-by-step guidance for users on how to use the ETAB API to run different downstream tasks on echocardiography data. The notebooks can be accessed online through this [link]( https://github.com/ahmedmalaa/ETAB/tree/main/notebooks). We hope that these modifications address your concern regarding the user-friendliness of the ETAB library interface.

---

### Official Review · Reviewer_3NVJ · 2022-07-28
**Pre-train possibilities in Echocardiography**

**Rating:** 4
**Confidence:** 4
**Correctness:** Yes
**Clarity:** Yes

**Strengths:**

- The paper involves three well-known public datasets: EchoNet, CAMUS and TMED.
- Adequate classification of adaptation tasks, including clinical prediction, view recognition, cardiac function estimation and cardiac structure identification.
- Comprehensive experimental setup with the use of popular methods such as ResNet50, U-Net etc
- Provided some basic insights on transfer and representation learning in Echocardiography.

**Weaknesses:**

- The paper presents a benchmarking suite to investigate the task adaptation of representation learning methods which is simplistic with applications in a very limited context.
- The adaptation benchmarks were proposed based on limited views/knowledge of the existing datasets, which might not be exhaustive and "unified" for tasks in Echocardiography.
- Whereas enabling adaptation of pre-trained models are good to have, how does ETAB impact the existing/new workflows in developing representation learning methods?
- Limited discussion on theoretical contributions and practical implications of the paper in real-world scenarios.


**Additional Feedback:**

NIL

**Documentation:**

Minimal

**Ethics:**

There is no new dataset.

**Relation To Prior Work:**

The paper did not discuss previous contributions.

**Summary And Contributions:**

The paper proposes a benchmark approach for representation learning of visual data in cardiac ultrasound. It provides 25 adaptation benchmarks in which a source task is performed on a source dataset, and then learned representation is employed for a target task on a target dataset. ETAB involves three datasets: EchoNet, CAMUS and TMED. The proposed benchmark suite captures a degree of adaptation on multiple datasets; nevertheless, the impact of this suite remains questionable.

---

> ### Author Response · Authors · 2022-08-29
> **Thanks for your feedback + Updates and changes in ETAB.**
>
> Thank you for your comments and feedback. Please refer to the combined response above for a summary of the updates in the ETAB software package and documentation conducted during the rebuttal period.
>
> The ETAB package provides the following features:
>
> - A unified API for loading and processing various publicly-accessible echocardiography datasets.
> - A comprehensive suite of (transfer learning and downstream) benchmark tasks tailored to echocardiography.
> - An easy-to-use API for creating modeling pipelines that supports a wide variety of backbone architectures and covers different types of downstream tasks of interest.
> - A unified evaluation protocol for pre-trained echocardiographic representations.
>
> We believe that the proposed benchmark tasks are very comprehensive and cover most views and prediction targets of interest in downstream analyses of echocardiography. The benchmark can be used for developing vision modeling pipelines or evaluating pre-trained representations, especially in scenarios where the target cohort contains a limited number of annotated samples. Moreover, we believe that ETAB can be a very powerful tool for academic research focused on analyzing echocardiograms.

---

> > ### Comment · Reviewer_3NVJ · 2022-08-29
> > **Reply to authors**
> >
> > I have read the authors' response and would like to keep my original rating.

---

> > > ### Comment · Reviewer_haAn · 2022-09-02
> > > **Can the reviewer clarify the perceived weaknesses?**
> > >
> > > Hi fellow reviewer,
> > >
> > > I'm having trouble understanding some of your outstanding concerns under "weaknesses", can you please clarify?
> > >
> > > Below, I'll go through a few of the weaknesses you list one by one and try to offer my own response / raise points of clarification
> > >
> > > ### Weakness 1
> > >
> > > > 1. The paper presents a benchmarking suite to investigate the task adaptation of representation learning methods which is simplistic with applications in a very limited context.
> > >
> > > My questions here are:
> > >
> > > * 1a: What exactly is "simplistic"?
> > >
> > > In my view, the paper looks at various transfer and pretraining scenarios, and covers a wide range of backbone architectures.
> > >
> > > * 1b: What exactly is the complaint about "very limited context"?
> > >
> > > Is the concern that the dataset is focused exclusively on echocardiography?
> > >
> > > If so, while I agree that this limits its scope and potential audience, I do think having an echocardiography benchmark like this is both extremely useful to practitioners in that sub area as well as potentially useful for representation learning in medical imaging more broadly.
> > >
> > > ### Weakness 2
> > >
> > > > The adaptation benchmarks were proposed based on limited views/knowledge of the existing datasets, which might not be exhaustive and "unified" for tasks in Echocardiography.
> > >
> > > Can you clarify more about why you see the proposed benchmark as "limited"?
> > >
> > > In my view, the paper covers a wide range of tasks relevant to echocardiography, from view classification to measurement to structure segmentation to diagnosis. Are there specific tasks that they missed that should be in here?
> > >
> > > It may be difficult to be truly exhaustive. I think they've covered the vast majority of common tasks in this space.
> > >
> > >
> > > ### Weakness 4
> > >
> > > > Limited discussion on theoretical contributions and practical implications of the paper in real-world scenarios.
> > >
> > > I would agree here that a bit more discussion of how the ETAB benchmark could *translate* to clinical applications would be helpful.
> > >
> > > But I don't think this itself is a reason to reject.
> > >
> > > ### RE previous work
> > >
> > > I'd argue that the paper did in fact situate their work with respect to previous contributions... see the part of their paper that says
> > >
> > > > Our benchmarking setup can be considered as the domain-specific analogue of the general benchmark for vision tasks de-
> > > veloped in [ 19 , 20].
> > >
> > > Perhaps there could have been a more *expanded* discussion, but I don't think I would say that the paper "did not discuss"

---

### Official Review · Reviewer_haAn · 2022-07-28
**Very promising work <strike>if poor reproducibility and lack of documentation can be addressed</strike>**

**Rating:** 7
**Confidence:** 4

**Strengths:**

Strengths are:

* Tackles an important problem for an exciting application area (echocardiography); could spur significant ML methods development that is *useful* for improving the efficiency and quality of patient care

* Broad coverage of tasks of interest within echocardiography, ranging from segmentation for structure, function estimation, view recognition, and diagnosis prediction

* Specific experiments that show benefits of pretraining on medical tasks (3DSeg8) vs generic images (ImageNet) are valuable

* Experiments that assess performance versus target set data size (as in Fig 3a) are quite valuable, esp. for thinking about combining public data with in-house private labeled sets that are small


**Weaknesses:**

**Update after author response on 2022-09-02**:

I have raised my score to an "accept", because

* W0, W2, and W3 below have been completely addressed.
* W1 has been mostly addressed (see my detailed comments in response to the authors for a remaining minor issue related to how CAMUS test data is used)

**Original review submitted in July 2022**

The key issues I see with the present paper are:

* W0: Confusing task definitions for a few tasks in Table 2
* W1: Missing target-only baseline in Fig 4 and Tab 4; makes hard to assess quality
* W2: Reproducibility is poor: key experimental design details / hyperparameters not available
* W3: Documentation not ready; seems to have many completely blank sections

See detailed subsections labeled "W__" below for elaboration on each of these weaknesses.

I think these are very addressable in the response period, and I look forward to engaging with the authors via discussion.

This work has lots of promise; I just want to emphasize that at present, the reproducibility and documentation really need to improve for this to be accepted.

**Additional Feedback:**

### 1) Any specific reason not to also include the Unity dataset of echos from the UK?

https://data.unityimaging.net/

Looks inter-operable with the datasets chosen: has PLAX views, measures EF, etc.

Not being aware of this resource until now is a fine answer, just seems natural way to extend the current work.

### 2) Can you suggest specific weights to assign to each task when doing the ETAB expectation in Eq 1?

I worry that specifying the ETAB score so generically has problems.
I'd rather see specific weights recommended for each task, or just define it in terms of uniform weights.
Most users of this benchmark will have far less domain knowledge about echocardiography than the authors, so their ability to pick sensible weights is questionable.



**Clarity:**


### W0: Confusing task definitions for a few tasks in Table 2

Part 1: How can you fix a view type when classifying views?

In Table 2, look at tasks with prefix CA44____. Decoding this means that we should use CAMUS data images from view "A4C" to classify views.... but shouldn't we not specify the type of view in advance if we want to classify it? Is this just a typo, and it should be CXX4____ instead?

Part 2: Need to clarify if any task is multi-class (> 2 categories)

For example, in TMED data (see Tab 1 of Huang et al), the AS task has 3 labels (no AS, mild-to-moderate AS, and severe AS). Is your AS classification task using all 3 labels, or a binary task? If using all 3, how do you compute an AUROC score... what do you average?


### Minor issues

Please address in revision, but no need to discuss in response period

- Should clarify that the EchoNet dataset used is from Stanford (current main paper doesn't specify the institution), and should clarify that it is "EchoNet Dynamic". This distinguishses from the recently released EchoNet LVH dataset: https://echonet.github.io/lvh/
- Under "Evaluation metrics" (line 187), mention that for all metrics, higher is better?
- Bolding policy in Tab 3 needs clarity.... why bold .893 but not .892 in first line?


**Correctness:**

I will raise some concerns about experimental quality here

### W1: Missing target-only baseline in Fig 4 and Tab 4; makes hard to assess quality

When evaluating whether a source task helps, it is important to consider the baseline of just "Training from scratch". While done in Fig 3, this is missing in Fig 4 and Tab 4. Makes it very difficult to know if any of this pretraining is worthwhile.

I'm particularly wondering about the view classification results in Fig 4, which appear to score in the 0.5-0.75 range for AUROC for the CXX4 target task (distinguish between A2C and A4C views). Previous publications indicate that view classification can be done much better: a 2018 study [*] reports 97% accuracy in a 15-way view classification task using 267 studies and a much simpler model (VGG16, not ResNet50). Perhaps there's an easy explanation (how many studies in target are available for fine-tuning?), but I think more explanation is needed here.


[*] Madani et al '18. Fast and accurate view classification of echocardiograms using deep learning.



**Documentation:**


### W2: Reproducibility is poor

Key experimental design details / hyperparameters not available

For example, in both appendix and github, I could not find:
- how data might be split among train/validation/test for each source task
- learning rate used for fine-tuning
- if any early stopping was used
- if any hyperparameter search is performed
- what weight was given to each task in Tab 2 when computing the ETAB scores in Tab 4

Also, please clarify whether the train/test splits used differ from original datasets recommended splits (if available).


### W3: Documentation not ready

The github repo seems to not be ready.

The main README referred to several sections that seem to be broken links / missing content entirely (as of 2022-07-28)

All of these raise **404 not-found errors when I tried them**

Datasets, accessibility and dataset tools
https://github.com/ahmedmalaa/ETAB/blob/main/documentation/data_access.md

Benchmark tasks
https://github.com/ahmedmalaa/ETAB/blob/main/documentation/benchmark_tasks.md

Leaderboard
https://github.com/ahmedmalaa/ETAB/blob/main/documentation/leaderboard.md


**Ethics:**

I don't have concerns. While they use medical images, the source datasets all are appropriately deidentified so there is no personal information and thus this isn't human subject research.


**Relation To Prior Work:**

While there have been several general-purpose vision benchmarks that asssess representations across tasks (see lit review in App B), this is the first one I'm aware of specific to echocardiography.

Some prior work has already shown that open-access echo datasets are inter-operable (e.g. train on Stanford, test on Unity). But the broad scope of this benchmark showing how many inter-operable tasks are possible is new to me.


**Summary And Contributions:**

The paper pursues the research question of how to evaluate representation learning for echocardiographic data (ultrasound images of the heart). This is tough because available open-access datasets, while valuable, each cover only a subset of the supervised tasks of interest and may not offer too many labeled examples for their chosen tasks.

The paper shows how to build representation learning benchmarks for many tasks of interest by remixing 3 open-access datasets: EchoNet from Stanford in California; CAMUS from University Hospital of St Etienne in France; and TMED from Tufts Medical Center in Massachusetts.

The first contribution is a suite of 25 specific "visual task adaption" benchmarks, listed in Table 2. Each involves a source and a target choice of dataset-view-label.

The second contribution is an evaluation protocol that helps grade overall representations, via an average across tasks and task-categories (Eq 1). The idea is to be able to assess the "overall usefulness" of a representation across tasks, via a single metric.

Experiments show how their contributions enable insights about whether pretraining helps (Fig 3), which deep architecture might be best (Tab 3), and whether pretraining on medical images is better than generic images (Fig 4).

---

> ### Author Response · Authors · 2022-08-29
> **Thank you for the excellent comments + Updates and changes in ETAB**
>
> We would like to thank the reviewer for the truly valuable comments and suggestions which helped us improve the quality of our proposed benchmarking setup. Throughout the rebuttal period, we have worked on addressing all your comments regarding the clarity of benchmark tasks’ definitions, comprehensiveness of the baseline models under consideration, and the need for detailed documentation and clear guidelines for reproducibility. Please refer to the combined response above for an overall summary of the improvements and changes that have been incorporated in the ETAB software library to address these issues. Below is a detailed point-by-point description of how we addressed the four areas of weakness you identified in our initial submission, along with pointers to the changes in the revised manuscript and [software package](https://github.com/ahmedmalaa/ETAB) that reflect our responses to the specific comments you raised in your review.
>
> &nbsp;
>
> ### **Addressing the four areas of weakness in the initial submission**
>
> &nbsp;
>
> **(W0) Improving clarity of task definitions**
>
> Please note that we have updated our task encoding system to improve its readability and make it easier to parse. We did so by changing the order of the characters within each code to put more emphasis on the task labels and views, and moved the dataset identifiers to occupy the last characters of the task code. Details on the modified task codes can be found in the [Section on benchmark tasks]( https://github.com/ahmedmalaa/ETAB/blob/main/docs/benchmark_tasks.md) in the [ETAB documentation]( https://github.com/ahmedmalaa/ETAB#documentation).
>
> *Part 1:* Thank you for pointing out to the typo in Table 2. You are correct; the source task code should be CXX4____ and not CA44____.  This typo has been fixed in the revised version of the manuscript.
>
> *Part 2:* Currently, all classification tasks in ETAB are limited to binary labels. In the TMED data setup, we predict whether a patient has AS without predicting the degree of severity, in which case the AUC-ROC is evaluated with respect to binary indicators of AS diagnoses. This has been clarified in the revised version of the paper (Lines 270-272). Extension to multi-label classification is straightforward.
>
> &nbsp;
>
> **(W1) View classification and “Train-from-scratch” baselines**
>
> There are several factors that could affect view classification performance, including whether classification is conducted using still images or videos, the quality and resolution of frames, presence of additional annotations (e.g. ECGs) that may cause label leakage, and the relatedness of the two views being classified. In the specific experiment you pointed out to, we believe that the fact that classification is conducted on top of still images with only 50 samples (CAMUS test data) used for the target task is the main reason why the performance does not exceed a 90% accuracy as reported in the 2018 study you referred to.
>
> We appreciate your suggestion regarding the incorporation of train-from-scratch baselines in all adaptation experiments. We believe that for these experiments to truly show the value of pre-training, one must vary the number of samples in the target task to examine few-shot performance as done in the experiment in Figure 3. For such a large number of experiments and baselines, visualizing these results in static figures and tables can be cumbersome. To address this, we are working on a leaderboard for adaptation experiments that is analogous to the [online ETAB leaderboard](https://github.com/ahmedmalaa/ETAB/blob/main/docs/leaderboard.md#etab-leaderboard) with an additional slider that allows tuning the number of target samples and evaluating the differential improvement in performance between pre-trained and trained-from-scratch baselines. We have also provided an easy-to-use API for running any benchmark experiment both in the adaptation and train-from-scratch modes while varying the number of training samples to allow users run customized versions of the experiment you suggested. Details are provided in the [ETAB model zoo documentation](https://github.com/ahmedmalaa/ETAB/blob/main/docs/benchmark_tasks.md#running-a-benchmark-experiment-out-of-the-box-demo-notebook).

---

> > ### Author Response · Authors · 2022-08-29
> > **Thank you for the excellent comments + Updates and changes in ETAB (Part 2)**
> >
> > **(W2) Documentation of experimental details + providing a reproducibility checklist**
> >
> > To improve the reproducibility of our experiments as well as future experiments conducted by users of the ETAB library we have provided the following:
> >
> > - **(a)** A list of default training and hyper-parameter settings for all experiments in ETAB is now added to Section B of the Supplementary Material.
> >
> > - **(b)** An easy-to-use API for instantiating datasets, composing modeling pipelines and running benchmark experiments out-of-the-box in a systematic fashion. Documentation of the list of all modeling pipelines supported by ETAB can be found [here]( https://github.com/ahmedmalaa/ETAB/blob/main/docs/benchmark_tasks.md#etab-model-zoo), and a description of the ETAB API for running benchmark experiments can be found [here]( https://github.com/ahmedmalaa/ETAB/blob/main/docs/benchmark_tasks.md#running-a-benchmark-experiment-out-of-the-box-demo-notebook).
> >
> > - **(c)** A number of [notebooks](https://github.com/ahmedmalaa/ETAB/tree/main/notebooks) demonstrating how to use ETAB to run any of the experiments in the paper as well as new custom experiments. These demo notebooks are provided through this [link](https://github.com/ahmedmalaa/ETAB/tree/main/notebooks).
> >
> > - **(d)** A [leaderboard](https://github.com/ahmedmalaa/ETAB/blob/main/docs/leaderboard.md) that keeps track of the best performing modeling pipelines with links to downloadable model weights and checkpoints to facilitate reproducibility is provided and maintained within the ETAB repository.
> >
> > - **(e)** Finally, we dedicate a complete Section within the ETAB documentation for issues related to reproducibility. In particular, we provide a [checklist](https://github.com/ahmedmalaa/ETAB/blob/main/docs/reproducibility_checklist.md) of items for users to consider when running a new benchmark and reporting a new contributed model to the ETAB leaderboard. This Section will also evolve over time as we receive feedback from users of the library.
> >
> > We hope that the improvements we introduced to the revised paper and the documentation for ETAB will help ensure the reproducibility of current and future experiments conducted using our framework.
> >
> > &nbsp;
> >
> > **(W3) Providing a complete documentation for the ETAB library + Demo Notebooks**
> >
> > A thorough [documentation](https://github.com/ahmedmalaa/ETAB#documentation) of the ETAB library with instructions on how to install our framework, download and process the datasets, and run any benchmark experiment out-of-the-box is provided in our GitHub [repo](https://github.com/ahmedmalaa/ETAB).
> >
> > The current version of the documentation is organized in 5 Sections that cover the following topics:
> >
> > [Section 1: Datasets, Accessibility and Data Processing Tools](https://github.com/ahmedmalaa/ETAB/blob/main/docs/data_access.md)
> >
> > * [Overview of the Supported Echocardiography Datasets](https://github.com/ahmedmalaa/ETAB/blob/main/docs/data_access.md#Datasets)
> > * [Instructions for Dataset Access](https://github.com/ahmedmalaa/ETAB/blob/main/docs/data_access.md#Instructions-for-dataset-access)
> > * [Data Loaders and Processing Tools](https://github.com/ahmedmalaa/ETAB/blob/main/docs/data_access.md#data-loaders-and-processing-tools-demo-notebook)
> >
> > [Section 2: ETAB Benchmark Suite and Model Zoo](https://github.com/ahmedmalaa/ETAB/blob/main/docs/benchmark_tasks.md)
> >
> > * [Benchmark Task Categorization and Encoding](https://github.com/ahmedmalaa/ETAB/blob/main/docs/benchmark_tasks.md#benchmark-task-categorization-and-encoding)
> > * [ETAB Model Zoo](https://github.com/ahmedmalaa/ETAB/blob/main/docs/benchmark_tasks.md#etab-model-zoo)
> > * [Running a Benchmark Experiment Out-of-the-Box](https://github.com/ahmedmalaa/ETAB/blob/main/docs/benchmark_tasks.md#running-a-benchmark-experiment-out-of-the-box-demo-notebook)
> >
> > [Section 3: The ETAB Evaluation Protocol](https://github.com/ahmedmalaa/ETAB/blob/main/docs/etab_protocol.md)
> >
> > * [Description of the Evaluation Protocol](https://github.com/ahmedmalaa/ETAB/blob/main/docs/etab_protocol.md#Description-of-the-Evaluation-Protocol)
> > * [Computing the ETAB Score](https://github.com/ahmedmalaa/ETAB/blob/main/docs/etab_protocol.md#computing-the-etab-score)
> >
> > [Section 4: Leaderboard and Benchmark Results](https://github.com/ahmedmalaa/ETAB/blob/main/docs/leaderboard.md)
> >
> > * [ETAB Leaderboard](https://github.com/ahmedmalaa/ETAB/blob/main/docs/leaderboard.md#etab-leaderboard)
> > * [How to contribute?](https://github.com/ahmedmalaa/ETAB/blob/main/docs/leaderboard.md#how-to-contribute)
> >
> > [Section 5: Reproducibility Checklist](https://github.com/ahmedmalaa/ETAB/blob/main/docs/reproducibility_checklist.md)
> >
> > Demo notebooks are provided in the relevant Sections to illustrate basic use cases of ETAB to users of the library. The next release of ETAB will also include Colab and Gradio (huggingface) demos to enable users to run pre-trained models from our leaderboard on their own echo data online, without the need to download the datasets or re-train the models.

---

> > > ### Author Response · Authors · 2022-08-29
> > > **Thank you for the excellent comments + Updates and changes in ETAB (Part 3)**
> > >
> > > &nbsp;
> > >
> > > ### **Addressing other comments and suggestions**
> > >
> > > &nbsp;
> > >
> > > **Enabling user-defined weights for the ETAB score**
> > >
> > > Currently, we use a uniform weighting scheme for all tasks within the ETAB score. The best weighting scheme will depend on which of the benchmark tasks are most relevant to the downstream use cases of interest for each user. To address your suggestion, we have implemented an option for manually setting the task weights when computing the ETAB score. Please refer to [Section 3](https://github.com/ahmedmalaa/ETAB/blob/main/docs/etab_protocol.md) in the [ETAB documentation](https://github.com/ahmedmalaa/ETAB/blob/main/docs) for details on how to customize the weighting function across tasks.
> > >
> > > &nbsp;
> > >
> > > **Incorporating the Unity dataset into the ETAB library**
> > >
> > > Thank you very much for making us aware of this great data resource. We have identified three new tasks that can be uniquely designed using the Unity dataset, all related to strain function analysis and estimating LV measurements beyond segmentation of the LV traces. These new benchmark tasks are listed in our [documentation of the ETAB benchmark suite]( https://github.com/ahmedmalaa/ETAB/blob/main/docs/benchmark_tasks.md#benchmark-task-categorization-and-encoding). Please refer to this link for more details. Unfortunately, we were not able to fully incorporate Unity into the current version of ETAB within the author discussion period, but we are planning to incorporate this new dataset in the next release.
> > >
> > > **All the minor issues you raised were addressed in the revised manuscript.**

---

> > > ### Comment · Reviewer_haAn · 2022-09-02
> > > **Thanks for the extensive new documentation and reproducibility details!**
> > >
> > > These thorough updates have addressed my concerns W2 and W3.

---

### Author Response · Authors · 2022-08-29
**Thanks to the reviewers! + Major changes and improvements in the ETAB frameworks**

We would like to thank all reviewers for their thoughtful and truly constructive comments. The excellent feedback we received from the reviewers helped us greatly improve the quality of our benchmarking framework in terms of experiment design, software design, API design and documentation. Throughout the rebuttal period, we have worked on addressing all the reviewers’ comments by creating a full-fledged software package for the ETAB benchmark, with additional modeling baselines that cover various SOTA backbone models of vision, in addition to an intuitive and user-friendly API that can enable users utilize ETAB in their own research. The ETAB package can be accessed online via this [link](https://github.com/ahmedmalaa/ETAB). In what follows, we summarize the major changes introduced to our submission along with links to these changes in the ETAB GitHub repository. Point-by-point responses to reviewer comments are also posted below.

&nbsp;


### **Summary of improvements in the ETAB framework based on reviewers’ comments**

&nbsp;

**(1)** ***A richer suite of backbone representations and task-specific heads***

We expanded the set of modeling pipelines supported in ETAB to cover 12 SOTA backbone visual representations that fall into two categories: convolutional neural networks and vision transformers. The list of all modeling pipelines currently supported by our framework can be found in the [ETAB model zoo](https://github.com/ahmedmalaa/ETAB/blob/main/docs/benchmark_tasks.md#etab-model-zoo). The newly incorporated backbones include SOTA CNN architectures (such as MobileNet and ConvNext) and transformer architectures such as Mix Transformer encoders, Multi-scale vision Transformer, and Pyramid Vision Transformer.

&nbsp;

**(2)** ***A full documentation of the ETAB software package***

A thorough [documentation](https://github.com/ahmedmalaa/ETAB#-documentation) of the ETAB framework is now available online. The documentation is meant to guide users on how to set up the ETAB package, download the echocardiography datasets, run standardized benchmark experiments out-of-the-box, and use the ETAB API to design custom modeling pipelines or new benchmark experiments. The current version of the ETAB documentation is organized in **5 Sections** that cover the following topics (click on the links below to view the respective Sections):

&nbsp;

[Section 1: Datasets, Accessibility and Data Processing Tools](https://github.com/ahmedmalaa/ETAB/blob/main/docs/data_access.md)

* [Overview of the Supported Echocardiography Datasets](https://github.com/ahmedmalaa/ETAB/blob/main/docs/data_access.md#Datasets)
* [Instructions for Dataset Access](https://github.com/ahmedmalaa/ETAB/blob/main/docs/data_access.md#Instructions-for-dataset-access)
* [Data Loaders and Processing Tools](https://github.com/ahmedmalaa/ETAB/blob/main/docs/data_access.md#data-loaders-and-processing-tools-demo-notebook)

&nbsp;

[Section 2: ETAB Benchmark Suite and Model Zoo](https://github.com/ahmedmalaa/ETAB/blob/main/docs/benchmark_tasks.md)

* [Benchmark Task Categorization and Encoding](https://github.com/ahmedmalaa/ETAB/blob/main/docs/benchmark_tasks.md#benchmark-task-categorization-and-encoding)
* [ETAB Model Zoo](https://github.com/ahmedmalaa/ETAB/blob/main/docs/benchmark_tasks.md#etab-model-zoo)
* [Running a Benchmark Experiment Out-of-the-Box](https://github.com/ahmedmalaa/ETAB/blob/main/docs/benchmark_tasks.md#running-a-benchmark-experiment-out-of-the-box-demo-notebook)

&nbsp;

[Section 3: The ETAB Evaluation Protocol](https://github.com/ahmedmalaa/ETAB/blob/main/docs/etab_protocol.md)

* [Description of the Evaluation Protocol](https://github.com/ahmedmalaa/ETAB/blob/main/docs/etab_protocol.md#Description-of-the-Evaluation-Protocol)
* [Computing the ETAB Score](https://github.com/ahmedmalaa/ETAB/blob/main/docs/etab_protocol.md#computing-the-etab-score)

&nbsp;

[Section 4: Leaderboard and Benchmark Results](https://github.com/ahmedmalaa/ETAB/blob/main/docs/leaderboard.md)

* [ETAB Leaderboard](https://github.com/ahmedmalaa/ETAB/blob/main/docs/leaderboard.md#etab-leaderboard)
* [How to contribute?](https://github.com/ahmedmalaa/ETAB/blob/main/docs/leaderboard.md#how-to-contribute)

&nbsp;

[Section 5: Reproducibility Checklist](https://github.com/ahmedmalaa/ETAB/blob/main/docs/reproducibility_checklist.md)

---

> ### Author Response · Authors · 2022-08-29
> **Thanks to the reviewers! + Major changes and improvements in the ETAB frameworks (Cont'd)**
>
> &nbsp;
>
> **(3)** ***A user-friendly API for running benchmark experiments out-of-the-box***
>
> We changed the design of ETAB to enable a more user-friendly API that allow users to easily customize modeling pipelines, design new benchmark tasks and incorporate new echocardiogram datasets. The API comprises a number of classes (*ETABdataset*, *ETABmodel* and *ETABBenchmark*) with unified built-in functionalities and methods that can enable users to capitalize on the features of the ETAB library in an abstract and modular manner. Details of these classes and the overall ETAB API are described throughout the [documentation Sections](https://github.com/ahmedmalaa/ETAB#-documentation).
>
> &nbsp;
>
> **(4)** ***Demo notebooks with step-by-step instructions on how to use ETAB***
>
> In addition to the thorough documentation, we have also provided several [demo notebooks](https://github.com/ahmedmalaa/ETAB/tree/main/notebooks) that provides users with step-by-step instructions on how to use the ETAB API and demonstrates the various features of the ETAB framework. These notebooks are referenced in the relevant Sections in the documentation and can be accessed online via this [link](https://github.com/ahmedmalaa/ETAB/tree/main/notebooks). The notebooks cover the following topics:
>
> * [How to load and process echocardiography data in ETAB?](https://github.com/ahmedmalaa/ETAB/blob/main/notebooks/Demo%201%20-%20ETAB%20Data%20Loading%20and%20Processing%20Tools.ipynb)
>
> * [How to run an ETAB benchmark experiment out-of-the-box?](https://github.com/ahmedmalaa/ETAB/blob/main/notebooks/Demo%202%20-%20ETAB%20Benchmark%20Tasks.ipynb)
>
> * [How to compute the ETAB score for a pre-trained representation?](https://github.com/ahmedmalaa/ETAB/blob/main/notebooks/Demo%203%20-%20Computing%20the%20ETAB%20score.ipynb)
>
> &nbsp;
>
> **(5)** ***A continuously updated leaderboard for backbone representations of echocardiograms***
>
> An [online leaderboard](https://github.com/ahmedmalaa/ETAB/blob/main/docs/leaderboard.md#etab-leaderboard) has been created to track the performance of the different backbone models and pre-trained weights with respect to the ETAB score (visit this [link](https://github.com/ahmedmalaa/ETAB/blob/main/docs/leaderboard.md#etab-leaderboard) to learn more). The leaderboard includes a breakdown of the ETAB score with respect to individual benchmark tasks and provides links to the pre-trained weights for the different models. The leaderboard provides a portal for users to contribute to ETAB by submitting their pre-trained models and evaluate these models systematically using the ETAB score.
>
> &nbsp;
>
> **(6)** ***A “reproducibility checklist” to improve reproducibility of ETAB score evaluation and benchmark experiments***
>
> Finally, we dedicate a complete Section within the ETAB documentation for issues related to reproducibility. In particular, we provide a [checklist]( https://github.com/ahmedmalaa/ETAB/blob/main/docs/reproducibility_checklist.md) of items for users to consider when running a new benchmark and reporting a new contributed model to the ETAB leaderboard. This Section will also evolve over time as we receive feedback from users of the library.

---

### Meta-Review · Area_Chair_Swai · 2022-09-07

**Recommendation:** Accept
**Confidence:** 3

**Metareview:**

The consensus among the reviewers was that this work covers an important topic and a broad number of tasks. There were concerns about the documentation, reproducibility, and accessibility of the dataset. But the authors have done a good job in addressing most of these concerns with documentation, an API, and example notebooks.

---

### Decision · Program_Chairs · 2022-09-16

Accept